# Use of the Chainchecker application: Uganda's experience during the 2022 Sudan Virus Disease outbreak

Rebecca Akunzirwe[1]*, Shannon Whitmer[2], Miles Stewart[3], Julie R. Harris[2], Mercy W. Wanyana[1], Sherry R. Ahirirwe[1], Alex R. Ario[1], Daniel Kadobera[1], Benon Kwesiga[1], Richard Migisha[1], Abraham Rajan[3], Nicole Stock[3], Julia Eng[3], John D. Klena[2], Trevor Shoemaker[2], Joel Montgomery[2], Mary Choi[2]

**1** Uganda Public Health Fellowship Program, Uganda National Institute of Public Health, Kampala, Uganda, **2** Centers for Disease Control and Prevention, Atlanta, Georgia, United States of America, **3** Applied Physic Laboratory, Johns Hopkins University, Baltimore, Maryland, United States of America

* akunzirwe@gmail.com

## Abstract

On September 20, 2022, the Uganda Ministry of Health declared an outbreak of Sudan Virus Disease (SVD). As the outbreak grew, it became imperative to quickly visualize and analyze chains of disease transmission. Determining epidemiological links between cases is critical for outbreak control as incorrect linkages may result in missed case detection and undetected disease transmission. We describe the Uganda Ministry of Health's experience using Chainchecker, a computer application designed to visualize and verify transmission chain data. To use Chainchecker, a line list documenting the epidemiological details associated with individual cases is uploaded to the application. To verify epidemiologic linkages, the application calculates the exposure windows for each case based on user-defined incubation periods and dates of symptom onset. If genetic sequencing data is available, Chainchecker can overlay genetic distance data on top of the epidemiologic data. Chainchecker can also provide visualizations of hospitalization data, which can highlight potential instances of nosocomial disease transmission. Using the Chainchecker application, the case investigation team was able to connect 11 previously unlinked cases to the larger chain of disease transmission. The use of the application also led to the identification and correction of transmission chain errors for 13 SVD cases and the identification of 5 potential instances of nosocomial transmission. The use of the Chainchecker application in Uganda during the 2022 SVD outbreak allowed the response teams to rectify critical errors in transmission chains. Countries prone to Ebola Disease (EBOD) outbreaks should consider incorporating Chainchecker as an element of EBOD preparedness and response.

## Introduction

On September 20, 2022, the Uganda Ministry of Health declared a Sudan Virus Disease (SVD) outbreak after a specimen collected from a 26-year-old male living in Madudu Subcounty, Mubende District, Central Uganda tested positive for Sudan virus (SUDV) by

**Data availability statement:** The data used in this study were obtained under strict confidentiality agreements with the Ugandan Ministry of Health and the Uganda Public Health Fellowship Program due to the sensitive nature of the Ebola outbreak data. To protect patient privacy and comply with national policies, these agreements prohibit public sharing of the data. However, de-identified datasets can be made available upon reasonable request. Investigators whose proposed use of the data has been approved by an independent review committee, designated for this purpose, may request access from the corresponding author with permission from the Uganda Ministry of Health and the Uganda Public Health Fellowship Program after the article's publication. Data requests may be sent to the Uganda National Institute of Public Health at info@uniph.go.ug, located at Lourdel Towers, 4th Floor, PO Box 7272, Kampala, Uganda. Tel: +256-312-800832.

**Funding:** The authors received no specific funding for this work.

**Competing interests:** The authors have declared that no competing interests exist.

reverse transcription polymerase chain reaction (RT-PCR) [1]. The outbreak spread to other sub-counties in Mubende District and to other districts, including the capital district of Kampala. In total, 164 confirmed and probable cases and 77 (47%) deaths were reported during the outbreak, which was declared over on January 11, 2023 [2].

Sudan virus (species *Orthoebolavirus sudanense*) is one of four viruses within the family *Filoviridae* that causes Ebola disease in humans. Early identification, isolation, and treatment of Ebola Disease (EBOD) cases are critical to reducing case fatality and stopping the outbreak. Active case finding, contact tracing, and epidemiologic linking of cases are key to the early identification of cases and require the timely and accurate collection, management, and analysis of epidemiologic data [3–5].

Previous EBOD outbreaks have been plagued by poor data management, including a lack of centralized data management systems, duplicate entries, and data inconsistencies [6]. Early in the 2022 SVD outbreak, transmission chains were maintained using paper-based systems and Microsoft PowerPoint. However, these were prone to error, inefficient, and difficult to analyze, especially as the outbreak expanded. To address these limitations, the Uganda SVD response team used Chainchecker, a computer application designed to visualize, curate, and verify disease transmission chain data [7]. We describe the Uganda Ministry of Health's experience using Chainchecker during the 2022 SVD outbreak.

## Methods

### Chainchecker application

Determining epidemiological links between confirmed cases is a critical component of EBOD outbreak control. Incorrect linkages may result in missed case detection and undetected disease transmission. However, the vast quantity of data available from different sources, coupled with the occurrence of data entry errors and input irregularities can make it challenging to discern epidemiological links. In addition, epidemiologic links are wholly dependent on patient or next-of-kin recall, which may be limited due to a variety of factors including stigma, fear, and illness severity. Genomic sequencing information can serve as an objective data point, but it has been a challenge to distil the information in a manner that is easy to interpret alongside the epidemiologic data. To this end, the U.S. Centers for Disease Control and Prevention (CDC) Viral Special Pathogens Branch (VSPB) and the Johns Hopkins Applied Physics Laboratory (APL) developed a new version of the Chainchecker application to visualize, curate, and verify disease transmission chain data using both epidemiological and genomic data [7].

To use Chainchecker, a line list documenting the epidemiological details associated with individual cases is uploaded to the application as an Excel or CSV file. The minimum required dataset consists of a case ID and illness onset date. To build transmission chains, a 'source' field

is required. This field identifies the known infecting source (represented by solid lines) or hypothetical sources, such as an animal or another individual (represented by dashed lines to indicate uncertainty). These connections provide an epidemiological link between cases. Additional information included in the epidemiological line listing can also be visualized such as epidemiologic classification (e.g., confirmed, probable), date of admission to a treatment unit, death dates, hospitalization data, and geographic information. As previously described, the application establishes a framework for the estimation and verification of epidemiologic links between cases [7]. One component of this framework is the calculation of the "exposure window", the time frame during which a patient was most likely infected with SUDV. The earliest exposure date is calculated by subtracting the maximum incubation period from the

date of symptom onset. The latest exposure date is calculated by subtracting the minimum incubation period from the date of symptom onset. Disease parameters such as the minimum incubation period and the maximum incubation period have preset values informed by previous outbreaks (these account for the health status of the infecting case i.e. whether they were in the dry phase, humid phase, or post-mortem exposure); however, these can be adjusted by the user [7].

These user-defined variables can be visualized in one of two ways: the 'chain of disease transmission' view or the 'healthcare facility' view. In the chain of disease transmission view, dates are shown on the horizontal x-axis, while case information is stacked vertically in order of illness onset date from earliest (top) to latest (bottom) (Fig 1).

Linkages between cases are represented by vertical lines. Users can set date limits for the chain of disease transmission that is displayed by adjusting the date range for the transmission chain. In the healthcare facility view, user-defined hospitalization data (date of admission, date of discharge, name of healthcare facility) are visualized as a function of a selected healthcare facility for a given date range or as a function of a selected patient for a given date range (Fig 2).

If a healthcare facility is selected, Chainchecker will display all the patients who transited through the selected healthcare facility during the selected date range. In this view, the unique IDs are shown on the vertical axis and time is shown on the horizontal axis. If a patient is selected, Chainchecker will display all the healthcare facilities the selected patient transited through during the selected date range. In this view, the names of the healthcare facilities appear on the vertical axis and time is shown on the horizontal axis.

Where available, genetic data can be added into the Chainchecker application using a text-based format. Genomes labeled with both laboratory and epidemiological identifiers are aligned [8], ambiguous or missing sites are removed, and the raw genetic distance is calculated using the dist.dna function in R library ape [9] and multiplied by the number of total analyzed nucleic acids [10]. Other nucleotide substitution models, such as TN93, can also be used to calculate genetic distance [11]. Before uploading into Chainchecker, the genetic

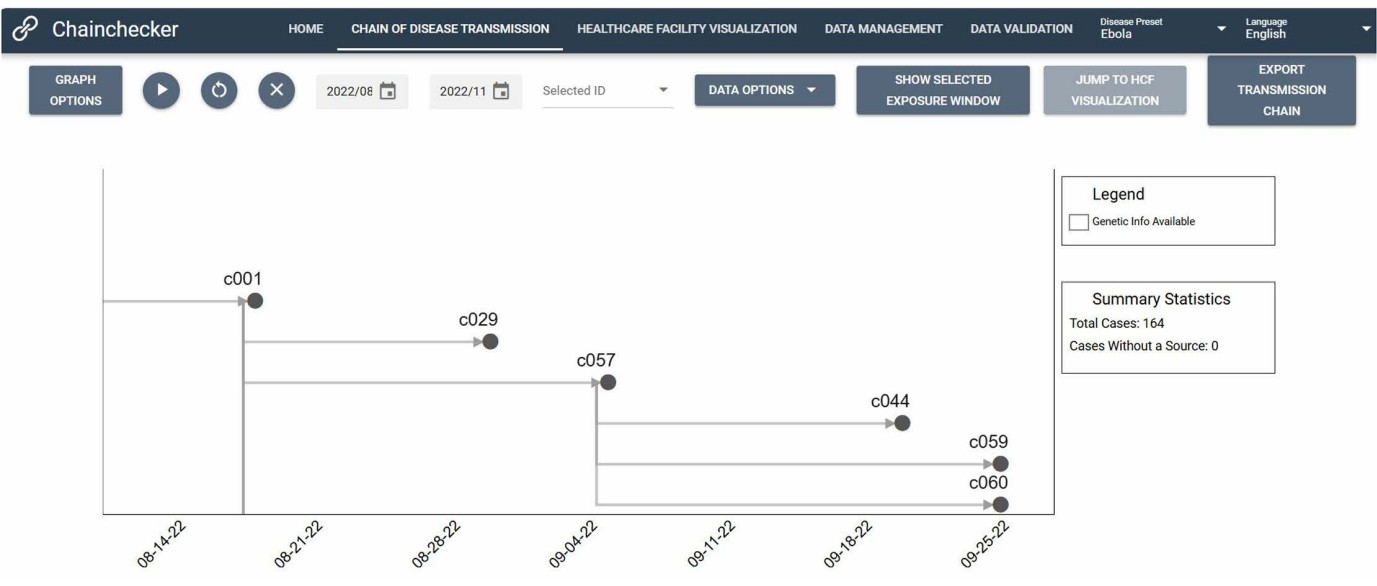

**Fig 1. Chain of disease transmission view.**

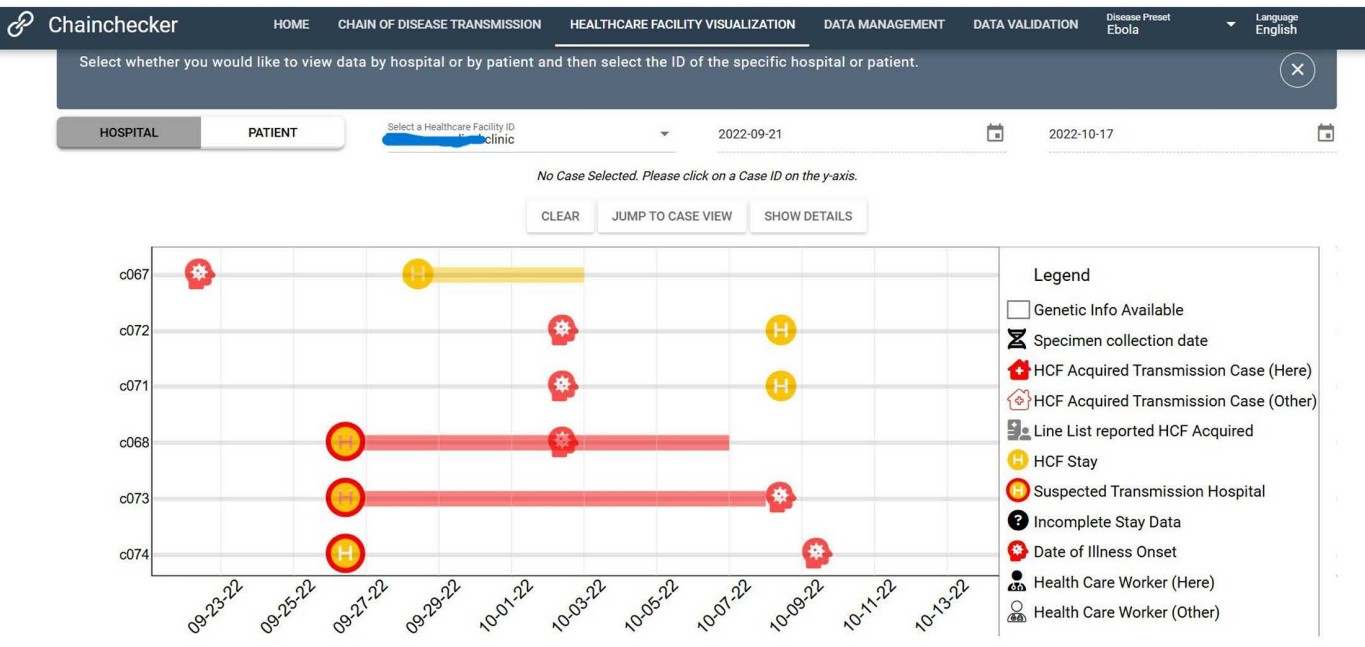

**Fig 2. Healthcare Facility View.**

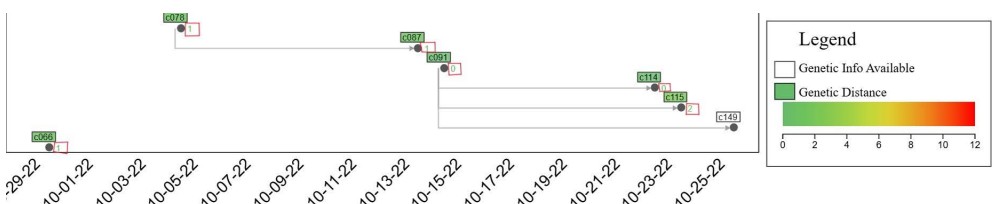

**Fig 3. Chainchecker illustration showing genetic distance between the selected individual (c149) and other cases (c078, c087, c091, c114, c115 and c066).** The nucleotide differences between the selected individual (c149) and each of the other cases are highlighted by red boxes.

distance matrix is converted into a long-based format and the lab identifiers are removed using in-house scripts [12]. The genetic distance between cases is depicted by a color scale and by numerical representation (Fig 3).

Cases with strong epidemiological support for direct transmission exhibited less than 4 nucleotide differences and using the substitution rates from the estimated time for a single mutation to emerge was 8.6 days [6.0–15.1 days, using 95% HPD lower and upper substitution rate estimates; $2.23 \times 10^{-3}$ ($1.274$–$3.179 \times 10^{-3}$ subs/site/year, 95% HPD)] [13].

Previously, the Chainchecker application was run as an R Shiny application either on the web or using R Portable [7]. The updated version described here has been re-written in Python and JavaScript as an offline application to allow for easier distribution (S1 Table).

## Ethical considerations

The Ministry of Health of Uganda gave the directive and approval to investigate this Sudan virus Disease outbreak. In agreement with the International Guidelines for Ethical Review

of Epidemiological Studies by the Council for International Organizations of Medical Sciences (1991) (39) and the Office of the Associate Director for Science, CDC/Uganda, it was determined that this activity was not human subject research and that its primary intent was public health practice or disease control activity (specifically, epidemic or endemic disease control activity). Verbal informed consent was obtained from the participants before the start of each interview as part of the epidemiological surveillance activities for EBOD; the interviews weren't undertaken as part of this specific study. Parental/legal guardian verbal informed consent was obtained on behalf of the children (less than 18 years old) before the start of each interview. The authors sought permission to conduct the study from the Ministry of Health, and Uganda Public Health Fellowship program. Data (accessed on 15/01/2023) did not contain individual personal identifiers as a way of ensuring confidentiality. This activity was reviewed by the CDC and was conducted consistent with applicable federal law and CDC policy. §

§See e.g., 45 C.F.R. part 46, 21 C.F.R. part 56; 42 U.S.C. §241(d); 5 U.S.C. §552a; 44 U.S.C. §3501 et seq

## Results

On November 8, 2022, in the middle of the SUDV outbreak, the Uganda Field Epidemiology Training Program (FETP) Team attended a hands-on training workshop on the use of the Chainchecker application. Led by CDC software developers, the training used de-identified outbreak data to demonstrate the functionalities of the application. On December 3, a refresher training was conducted for the FETP team. In total, 45 individuals were trained on the application: 34 FETP trainees, 6 FETP staff, and 5 CDC staff involved in active case finding and contact tracing. Using Chainchecker, the team was able to recreate chains of disease transmission that were previously hand-drawn (Fig 4).

Leveraging Chainchecker's analysis tools, the team was able to connect 11 previously unlinked cases to the larger chain of disease transmission and identify and correct errors in the epidemiologic linkages for 13 cases. The use of the application also led to the identification of 5 potential instances of nosocomial transmission. We describe three illustrative examples below.

### Example 1

C 065, a 7-year-old male from Mubende District, Bagezza subcounty developed headache and general body weakness on September 24, 2022. On September 25, he was treated at Health Facility Y for malaria and was discharged on September 27. On September 29, he returned to Health Facility Y with general body weakness and nausea. On October 2, a blood sample was collected and confirmed positive for SUDV by RT-PCR. C065 was the first case of SUDV from Bagezza sub-county, where no other cases had been reported. Although case investigation data suggested that C065 was likely exposed to SUDV at Health Facility Y, the team was not able to identify the source of C065's infection using hand-drawn disease transmission chains. Using Chainchecker's healthcare visualization view, the team was able to identify three other cases (C033, C005, and C010) who were at Health Facility Y around the same time as C065. Of these, only C033 had been at Health Facility Y during the likely exposure window for C065 (September 9-19, 2022), and this patient represented the most likely source of C065's infection. Subsequent genetic sequencing of specimens collected from C033 and C065 showed the genomes to be highly genetically similar (3 nucleotide differences out of 18,875 bases), providing additional support for this epidemiologic linkage.

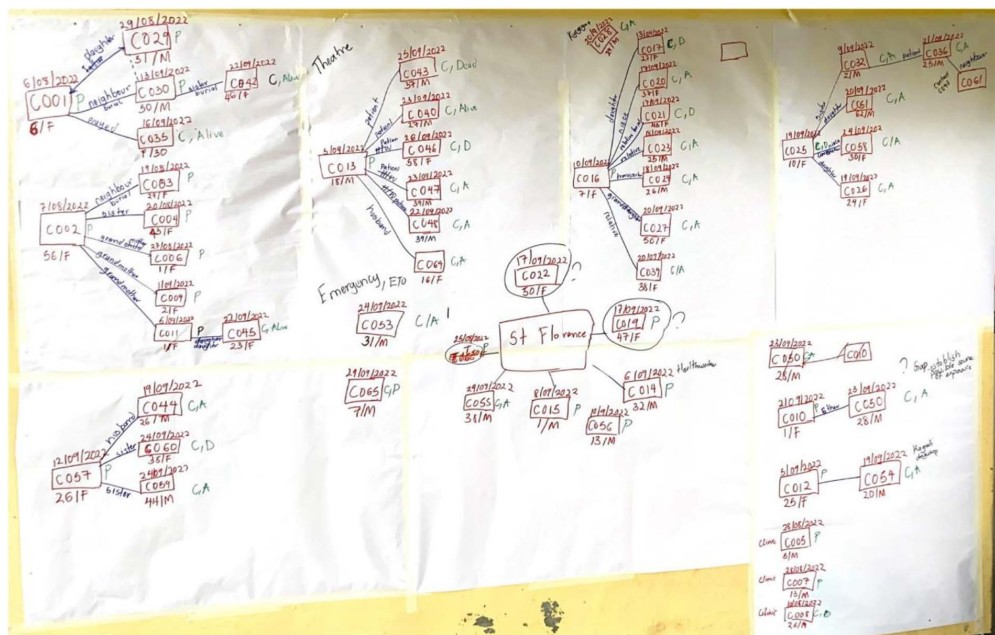

**Fig 4. Hand-drawn diagram of the disease transmission chain.**

## Example 2

C016 was a 6-year-old female from Mubende District. During September 4–5, she was treated for an ear infection at Health Facility X as an outpatient. On September 8, she developed a fever and general body weakness and was treated at Health Facility Y. On September 12, she died at home with bleeding from the nose and mouth. Initial case investigations linked C008, a 26-year-old male from Mubende District, to C016 as her likely source of infection as he had also been at Health Facility X during August 29–September 4. Utilizing Chainchecker's chain of disease transmission view, the team was able to visualize all the persons who were ill during C016's exposure window. The visualization made it evident that both C016 and C008 had symptom onset on September 08, 2022, making it unlikely that C008 was the source of infection for C016. The visualization also identified six other cases who were ill during C016's exposure period (C001, C003, C004, C005, C006 and C029). Further investigations revealed that of these six cases, C016 had only been in contact with C004, a 43-year-old female from Mubende District, who had been receiving treatment from Health Facility X for a malaria-like illness that was later determined to be due to SUDV infection. No genetic data was available from C0016 and C004, but by visualizing all of the cases and their metadata from the outbreak, the team was able to identify the likely source of C004's SUDV infection using the Chainchecker application.

## Example 3

Case C077, a 25-year-old female, was the wife and caretaker to confirmed case C069. On October 12, 2022, she went into pre-term labor at Health Facility Q and delivered a live infant. A specimen collected from C077 confirmed SUDV infection by RT-PCR on October 13, and her presumed source was her husband C069. To confirm this direct epidemiologic link, the team used Chainchecker to overlay genetic sequencing data obtained from C077 and C069. By doing this, the team discovered that there were 7 nucleotide differences between the sequences

obtained from C077 and C069, suggesting that direct transmission from husband to wife was unlikely. Instead, the genetic data indicated that C077's sequence was more closely related to that of her brother-in-law, C070 (nucleotide difference=2). C077's exposure window overlapped with the time period when C070 was ill, adding evidence to a likely transmission.

## Discussion

Initially, in response to the EBOD outbreak in Uganda in 2022, transmission chains were manually drawn. However, this proved cumbersome, time-consuming and hindered the efficient sharing of data. Correcting errors in the hand-drawn transmission chains required re-thinking and drawing the transmission chain from scratch [14,15]. While there are other software applications that can generate chains of disease transmission, the transmission chains are not mapped against time; are unable to incorporate and overlay additional metadata (such as multiple hospital visits) and genetic sequencing data on top of the epidemiologic data; and do not include functionality that can check for and highlight inconsistent epidemiologic links [7,16].

When the team converted from manually drawn chains of transmission to Chainchecker, they were able to correct 24 errors in the chains of disease transmission (11 unlinked cases and 13 erroneously linked cases), which accounted for 15% of all confirmed and probable cases reported during the outbreak. This finding highlights the limitations of determining chains of disease transmission by patient interviews alone. Both unlinked cases and erroneously linked cases are concerning for undetected transmission and can prolong outbreaks [17]. By bringing to bear checks for incubation periods, exposure windows, and genetic sequencing, Chainchecker was able to guide epidemiologists in rapidly identifying and correcting errors in disease transmission chains.

Chainchecker was instrumental in enabling rapid visualization, analysis, and verification of transmission chains and contributed to the successful management of epidemiological and genetic data during this outbreak. The user interface was intuitive, easy to navigate, and accessible to a wide range of users with varying levels of technological proficiency [7,18]. Additionally, the integration of data from various sources, including genomic sequence data, generated a more comprehensive view of transmission dynamics than could be visualized from paper-based hand drawings [19,20].

During Ebola disease outbreaks, patients often seek care at progressively higher levels of medical care before they are identified and confirmed with EBOD. Thus, it is not uncommon to have several EBOD patients admitted to the same tertiary care/referral hospital prior to confirmation. In these situations, there may be an inclination to erroneously attribute the mode of transmission for these cases as nosocomial.

Visualizing health facility visits, a feature unique to Chainchecker played a crucial role in identifying possible nosocomial infections. Visualizing the temporal movement of cases within healthcare facilities, while taking into account exposure windows, enabled the case investigation team to better understand and interpret exposure patterns and identify likely instances of nosocomial infection while discounting others.

Although Chainchecker is a valuable tool for real-time outbreak response, its ability to capture uncertainty is limited primarily to exposure windows defined by minimum and maximum incubation periods. It does not account for uncertainties in reported dates of symptom onset, exposure, or infectious periods, which may arise from recall bias or incomplete data. Future enhancements could address these limitations by incorporating probabilistic modelling or confidence intervals for key inputs and integrating additional metadata. Despite these constraints, Chainchecker remains a useful complement to other epidemiological and genomic data, enabling more effective outbreak response.

## Conclusion

Chainchecker was a critical tool for the verification of transmission chains during the 2022 SVD outbreak due to its automated visualizations of a complex dataset. The successful management of epidemiological and genetic data are important components of outbreak control because missed cases can lead to undetected and ongoing disease transmission. Countries that are prone to EBOD outbreaks should consider incorporating Chainchecker as an element of EBOD preparedness and response.

Since the software was used in Uganda, the user interface has been translated into Spanish and French. Additionally, as the application utilizes user-defined variables and adjustable preset values, it is easily modifiable for use during other infectious disease outbreaks. Chainchecker is designed as a versatile tool that can be adapted for use in tracking transmission dynamics of various infectious diseases, including Mpox, during sustained human-to-human transmission. The tool relies on user-defined parameters, such as incubation periods, dates of symptom onset, and periods of infectiousness, which can be customized to reflect the epidemiological characteristics of any disease.

## Supporting Information

**S1 Table. Updates made to Chainchecker capabilities since the initial R Shiny version.** (DOCX)

## Author contributions

**Conceptualization:** Rebecca Akunzirwe, Shannon Whitmer, Miles Stewart, Mary Choi.

**Data curation:** Rebecca Akunzirwe, Miles Stewart.

**Formal analysis:** Rebecca Akunzirwe, Shannon Whitmer, Miles Stewart, Julie R. Harris, Mercy W. Wanyana, Mary Choi.

**Investigation:** Rebecca Akunzirwe, Julie R. Harris, Sherry R. Ahirirwe, Alex R. Ario, Daniel Kadobera, Benon Kwesiga, Richard Migisha, Mary Choi.

**Methodology:** Rebecca Akunzirwe, Shannon Whitmer, Miles Stewart, Sherry R. Ahirirwe, Alex R. Ario, Abraham Rajan, Nicole Stock, Julia Eng, Mary Choi.

**Software:** Miles Stewart, Abraham Rajan, Nicole Stock, Julia Eng, John D Klena, Trevor Shoemaker, Joel Montgomery, Mary Choi.

**Supervision:** Shannon Whitmer, Mary Choi.

**Validation:** Shannon Whitmer, Benon Kwesiga, Richard Migisha, John D Klena, Trevor Shoemaker, Joel Montgomery.

**Visualization:** Rebecca Akunzirwe, Julie R. Harris, Mary Choi.

**Writing – original draft:** Rebecca Akunzirwe, Julie R. Harris, Mary Choi.

**Writing – review & editing:** Rebecca Akunzirwe, Shannon Whitmer, Miles Stewart, Julie R. Harris, Mercy W. Wanyana, Sherry R. Ahirirwe, Alex R. Ario, Daniel Kadobera, Benon Kwesiga, Richard Migisha, Abraham Rajan, Nicole Stock, Julia Eng, John D Klena, Trevor Shoemaker, Joel Montgomery, Mary Choi.

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
