## [Decision Letter · Decision Letter 0]

4 Nov 2024

PGPH-D-24-01253

Use of the Chainchecker application: Uganda’s experience during the 2022 Ebola Disease outbreak

Dear Dr. Akunzirwe,

Thank you for submitting your manuscript to PLOS Global Public Health. After careful consideration, we feel that it has merit but does not fully meet PLOS Global Public Health’s publication criteria as it currently stands. Therefore, we invite you to submit a revised version of the manuscript that addresses the points raised during the review process.

We look forward to receiving your revised manuscript.

Kind regards,

Flavio Finger

Guest Editor

Journal Requirements:

1. Please provide additional details regarding participant consent. In the ethics statement you have stated that verbal informed consent was obtained from participants before the start of each interview. Please can you clarify if the participants were consenting to participation in epidemiological surveillance or this specific research study.

Please can you also clarify whether the interviews took place as part of epidemiological surveillance and whether the data from these were anonymised prior to you accessing it? Or were the interviews conducted by you directly as part of this study.

2. Please provide separate figure files in .tif or .eps format.

Additional Editor Comments (if provided):

Reviewers' comments:

Reviewer's Responses to Questions

**Comments to the Author**

1. Does this manuscript meet PLOS Global Public Health’s publication criteria ? Is the manuscript technically sound, and do the data support the conclusions? The manuscript must describe methodologically and ethically rigorous research with conclusions that are appropriately drawn based on the data presented.

Reviewer #1: Yes

Reviewer #2: Yes

Reviewer #3: Yes

Reviewer #4: Yes

2. Has the statistical analysis been performed appropriately and rigorously?

Reviewer #1: N/A

Reviewer #2: Yes

Reviewer #3: Yes

Reviewer #4: Yes

3. Have the authors made all data underlying the findings in their manuscript fully available (please refer to the Data Availability Statement at the start of the manuscript PDF file)?

Reviewer #1: Yes

Reviewer #2: Yes

Reviewer #3: No

Reviewer #4: No

4. Is the manuscript presented in an intelligible fashion and written in standard English?

Reviewer #1: Yes

Reviewer #2: Yes

Reviewer #3: Yes

Reviewer #4: Yes

5. Review Comments to the Author

Reviewer #1: Overall, this is a solid paper on a very useful application, with clear use cases. This tool will make epidemic investigations easier and faster, and the combination of genomic and epidemiological data in particular is a very important innovation.

I have some minor comments

• It sounds like chainchecker is used on a computer with CSV data uploaded – are there any plans to make it a phone app which you can input data directly into?

• An example of what would go in the “source” field would be helpful – is that a location, or a known infecting case?

• There should be a citation for ape on line 120

• I’m not clear on how other nucleotide substitution models would be included, given that it sounds like the raw distance is what’s used? Some more detail on how genetic distance is calculated would be helpful.

• What software is used to align sequences?

• How are indels taken into account in terms of genetic distance? Ebola viruses have intergenic regions, and so indels may be real.

• How do you exclude poorly sequenced data? Or in general is there any quality control in the application itself, or are you relying on good genomic data and associated metadata going in? Without any quality control, there may be transmissions ruled out that are real epidemiological connections.

• Does the epidemiological connection have to be direct? Or does it just mean they are part of the same chain?

• NB The figures are very low resolution so it’s hard to assess them.

Reviewer #2: This is a very interesting paper explaining how an innovative application can help to determine epidemiological link between Ebola cases which is the key to adequately response and control an Ebola Virus Disease outbreak.

However, here are a few comments :

1. How does the health status of a confirmed case (dry phase, humid phase or dead body) impact the calculation of the incubation knowing that the risk to develop the disease may not be the same when contact occurs at those different phases of the disease?

2. Figure 1a and 1b are not readable.

3. To what extent Chainchecker can be used in different diseases such Mpox, in case of sustained human-to-human transmission?

Reviewer #3: The authors present a description of the use of the application ChainChecker for a recent disease outbreak in Uganda. Whilst the application has been previously described, the authors present an updated version and, of particular note, validate its use with genomic data. This highlights the benefits of clear visualisations for real time outbreak response. Though, the current work suffers from one of the same issues as the first tool: the tool captures uncertainty only through a range on the exposure window rather than in other areas such as the reported dates. However, as shown, coupled with other data sources and expertise of those conducting the outbreak response, it can contribute to the suite of activities.

Major notes: It would be good to expand on the limitation of uncertainty in the discussion ie. that it is only captured through the exposure window.

Minor notes: the references do not appear to line up correctly. Often reference 8 is mentioned for chainchecker but this refers to a different outbreak of Hepatitis rather than reference 7 which desciribes the original chainchecker tool.

Reviewer #4: In the title, change Ebola virus disease into Sudan virus disease, as this latter is most cited in the manuscript.

Introduction

1) Maybe authors can use standard abbreviations such as EVD instead of EBOD

2) Authors could lay more on inconsistencies created by the use of previous data reporting system to show the associated gaps, then the need to fully address these through the chainchecker application.

Methods

1) Line 87-89: how maximum and minimum incubation periods were they set for all confirmed-cases? Were they fixed or varying from one case to another? How did the authors determine them throughout the study? Can authors can lay more on that.

2) Line 95-97 is unclear. Would authors meant “stacked vertically”? That would more sense for the full sentence…

3) Line 142: correct this “de-identified data outbreak data”

4) Lines 144-146: look like results

5) Ethical considerations section can be summarized.

Discussion

1) The discussion is short and not consistent enough to support findings of this study.

2) Provide an in-depth discussion of key findings

3) Provide study limitations

4) Is it possible to extend the use of this tool to other diseases, such as mpox? (Lay a little bit more on that, not only one or two sentences).

6. PLOS authors have the option to publish the peer review history of their article (what does this mean? ). If published, this will include your full peer review and any attached files.

**Do you want your identity to be public for this peer review?** For information about this choice, including consent withdrawal, please see our Privacy Policy .

Reviewer #1: No

Reviewer #2: **Yes: ** Placide Mbala-Kingebeni

Reviewer #3: No

Reviewer #4: No

---

## [Editor Report · Decision Letter 1]

10 Feb 2025

Use of the Chainchecker application: Uganda’s experience during the 2022 Sudan Virus Disease outbreak

PGPH-D-24-01253R1

Dear Ms. Akunzirwe,

We are pleased to inform you that your manuscript 'Use of the Chainchecker application: Uganda’s experience during the 2022 Sudan Virus Disease outbreak' has been provisionally accepted for publication in PLOS Global Public Health.

Best regards,

Flavio Finger

Guest Editor